# HorNet: Efficient High-Order Spatial Interactions with Recursive Gated Convolutions

Yongming Rao[1*]  Wenliang Zhao[1*]  Yansong Tang[1]
Jie Zhou[1†]  Ser-Nam Lim[2†]  Jiwen Lu[1†]
[1]Tsinghua University  [2]Meta AI

## Abstract

Recent progress in vision Transformers exhibits great success in various tasks driven by the new spatial modeling mechanism based on dot-product self-attention. In this paper, we show that the key ingredients behind the vision Transformers, namely input-adaptive, long-range and high-order spatial interactions, can also be efficiently implemented with a convolution-based framework. We present the Recursive Gated Convolution ($g^n$Conv) that performs high-order spatial interactions with gated convolutions and recursive designs. The new operation is highly flexible and customizable, which is compatible with various variants of convolution and extends the two-order interactions in self-attention to arbitrary orders without introducing significant extra computation. $g^n$Conv can serve as a plug-and-play module to improve various vision Transformers and convolution-based models. Based on the operation, we construct a new family of generic vision backbones named HorNet. Extensive experiments on ImageNet classification, COCO object detection and ADE20K semantic segmentation show HorNet outperform Swin Transformers and ConvNeXt by a significant margin with similar overall architecture and training configurations. HorNet also shows favorable scalability to more training data and a larger model size. Apart from the effectiveness in visual encoders, we also show $g^n$Conv can be applied to task-specific decoders and consistently improve dense prediction performance with less computation. Our results demonstrate that $g^n$Conv can be a new basic module for visual modeling that effectively combines the merits of both vision Transformers and CNNs. Code is available at https://github.com/raoyongming/HorNet.

## 1 Introduction

Convolutional neural networks (CNN) have driven remarkable progress in deep learning and computation vision since the introduction of AlexNet [31] in the last decade. There are quite a few nice properties of CNNs making them naturally suitable for a wide range of vision applications. Translation equivariance introduces useful inductive biases to major vision tasks and enables transferability across different input resolutions. The highly optimized implementation makes it efficient on both high-performance GPUs and edge devices. The evolution of architectures [32, 31, 49, 50, 22, 24, 51] further increases its popularity on various vision tasks.

The emergence of Transformer-based architectures [16, 52, 42] greatly challenges the dominance of CNNs. By combining some successful designs in CNN architectures and the new self-attention mechanism, vision Transformers have shown leading performance on various vision tasks such as image classification [12, 42, 48], object detection [70, 41], semantic segmentation [6, 8] and video understanding [64, 18]. *What makes vision Transformers more powerful than CNNs?* Some efforts have been made to improve the CNN architectures by learning from the new designs in vision

---

*Equal contribution.  †Corresponding authors.

36th Conference on Neural Information Processing Systems (NeurIPS 2022).

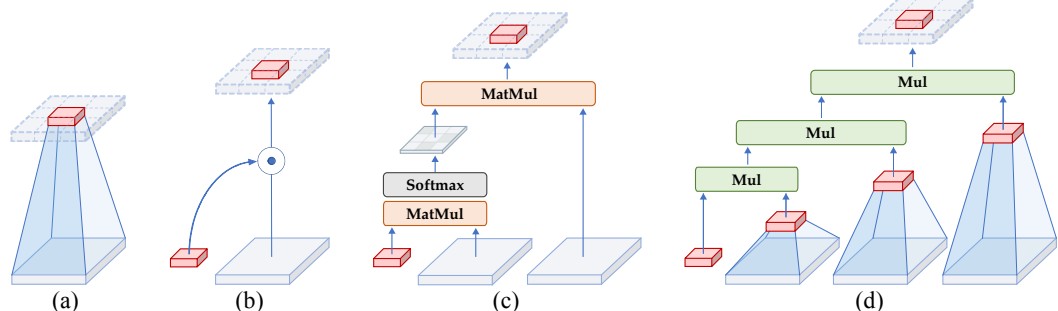

Figure 1: **Illustration of our main idea.** We show representative spatial modeling operations that perform different orders of interactions. In this paper, we focus on studying *explicit* spatial interactions between a feature (red) and its neighboring region (light gray). (a) The standard convolution operation does not explicitly consider the spatial interaction. (b) Dynamic convolution [28, 4] and SE [25] introduce the dynamic weights to improve the modeling power of convolutions with extra spatial interactions. (c) The self-attention operation [56] performs two-order spatial interactions with two successive matrix multiplications. (d) $g^n$Conv realizes arbitrary-order spatial interactions using a highly efficient implementation with gated convolutions and recursive deigns.

Transformers. [43] presents a thorough study to adopt the *meta architecture* of vision Transformer to improve CNNs and proposes to use a *large 7×7 kernel* to construct a modern CNN. [46] and [14] propose to use even *larger kernels* to learn long-range relations with global filters and up to 31×31 convolutions, respectively. [20] shows that the *input-adaptive weights* play a key role in vision Transformers and achieve similar performance with Swin Transformers with dynamic convolutions [4, 28]. However, the effectiveness of dot-product self-attention in vision tasks has not been analyzed from the prospective of *high-order spatial interactions*.

While there exists complex and often high-order interactions between two spatial locations in a deep model due to the non-linearity, the success of self-attention and other dynamic networks suggests that the *explicit* and *high-order* spatial interactions introduced by the architectural designs are beneficial to improving the modeling power of vision models. As illustrated in Figure 1, the plain convolution operation does not explicitly consider the spatial interactions between a spatial location (*i.e.*, the red feature) and its neighboring region (*i.e.*, the light gray region). Enhanced convolution operations like dynamic convolution [4, 28, 20] introduce explicit spatial interaction by generating dynamic weights. The dot-product self-attention operation in Transformers [56] consists of two successive spatial interactions by performing matrix multiplication among queries, keys and values. The trend of the basic operations for visual modeling indicates that the network capacity can be improved by increasing the order of spatial interactions.

In this paper, we summarize that the key ingredient behind the success of vision Transformers is the new way of spatial modeling with *input-adaptive*, *long-range* and *high-order* spatial interactions performed by the self-attention operation. While previous work has successfully migrated the meta architecture [43, 20, 46, 14], input-adaptive weight generation strategy [20] and large-range modeling ability [46, 14] of vision Transformers to CNN models, a higher-order spatial interaction mechanism has not been studied. We show that all the three key ingredients can be efficiently implemented using a convolution-based framework. We propose the Recursive Gated Convolution ($g^n$Conv) that performs high-order spatial interactions with gated convolutions and recursive deigns. Instead of simply imitating the successful designs in self-attention, $g^n$Conv has several extra favorable properties: 1) *Efficient*. The convolution-based implementation avoids the quadratic complexity of self-attention. The design that progressively increases the channel width during performing spatial interactions also enables us to achieve higher-order interactions with bounded complexity; 2) *Extendable*. We extend the two-order interaction in self-attention to arbitrary orders to further improve the modeling power. Since we do not make assumptions on the type of spatial convolution, $g^n$Conv is compatible with various kernel size and spatial mixing strategies like [46, 14]; 3) *Translation-equivariant*. $g^n$Conv fully inherits the translation equivariance of the standard convolution, which introduces beneficial inductive biases to major vision tasks and avoids the asymmetry brought by local attention [42, 34].

Based on $g^n$Conv, we construct a new family of generic vision backbones named HorNet. We conduct extensive experiments on ImageNet classification [13], COCO object detection [38] and ADE20K semantic segmentation [71] to verify the effectiveness of our models. With the same 7×7 kernel/window and similar overall architecture and training configurations, HorNet outperforms Swin

and ConvNeXt by a large margin on all tasks at different levels of complexity. The gap can be further enlarged by using a global kernel size [46]. HorNet also shows favorable scalability to more training data and larger model size, attaining 87.7% top-1 accuracy on ImageNet, 57.9% mIoU on ADE20K val and 59.2% bounding box AP on COCO val with ImageNet-22K pre-training. Apart from applying $g^n$Conv in visual encoders, we further test the generality of our designs on task-specific decoders. By adding $g$Conv to the widely used feature fusion model FPN [36], we develop HorFPN to model the high-order spatial relationships of features from different hierarchical levels. We observe that HorFPN can also consistently improve various dense prediction models with lower computational costs. Our results demonstrate that $g^n$Conv can be a promising alternative to self-attention for visual modeling and effectively combine the merits of both vision Transformers and CNNs.

## 2 Related Work

**Vision Transformers.**   The Transformer architecture [56] is originally designed for the natural language processing tasks. Since Dosovitskiy *et al*. [16] show that vision models constructed only by the Transformer blocks and a patch embedding layer can also achieve competitive performance to CNNs, many new models have been proposed to modify the Transformer-based architecture and make it more suitable for various vision tasks [42, 58, 60, 9, 66, 55]. Different from the original designs in [16], state-of-the-art vision Transformers usually utilize a CNN-like hierarchical architecture and change the global self-attention among all patches to local self-attention to avoid the quadratic complexity. In this paper, we follow the overall architecture of the previous hierarchical vision Transformers [42] and replace the self-attention sub-layer with our proposed $g^n$Conv to fairly compare with the previous Transformer-based models.

**Convolution-based models.**   Inspired by the recent success of vision Transformers, several papers propose to adopt the Transformer-style architecture and spatial convolutions with a large kernel size to improve the performance of CNNs. Han *et al*. [20] replace the window self-attention in Swin Transformers with large-kernel dynamic convolutions and achieve better performance. GFNet [46] proposes to perform the global spatial interactions like vision Transformers with global filters in the frequency domain, which are equivalent to depth-wise convolutions with a global kernel size and circular padding. ConvNeXt [43] thoroughly analyzes the designs in recent vision Transformers and presents a strong convolutional model with $7\times7$ depth-wise convolutions. RepLKNet [14] explores CNN models with very large kernels (up to $31\times31$), showing good scalability as vision Transformers. VAN [19] and FocalNet [65] use gated convolutions to perform input-adaptive attention and adopts large-kernel dilated convolutions and multiple successive $3\times3$ convolutions respectively to produce the weights. Previous work focuses on the meta architecture [67], large-kernel designs and input-adaptive weights to improve CNNs by learning from vision Transformers. In this paper, we offer a new perspective of high-order spatial attention to analyze the merits of vision Transformers. We show that the proposed HorNet that combines the advantages of both CNNs and vision Transformers is a better architecture for various vision tasks.

**Hybrid models.**   Combining vision Transformers and CNNs to develop hybrid architectures is a new direction in various visual recognition problems. Recently, several efforts have been made to integrate the two types of blocks into a unified model with a sequential [12, 29, 68, 63] or parallel [45, 11] design. Many enhanced vision Transformers also use lightweight convolutions in the basic building block to efficiently capture neighboring patterns [15, 60, 17] or relax the quadratic complexity of self-attention [9, 58, 18]. Different from these hybrid models, we aim to develop a self-attention free model while combining the favorable properties of both vision Transformers and CNNs.

## 3 Method

### 3.1 $g^n$Conv: Recursive Gated Convolutions

In this section, we will present $g^n$Conv, an efficient operation to achieve long-term and high-order spatial interactions. The $g^n$Conv is built with standard convolutions, linear projections and element-wise multiplications, but has a similar function of input-adaptive spatial mixing to self-attention.

**Input-adaptive interactions with gated convolution.**   Recent success in vision Transformers mainly depends on the proper modeling of the spatial interactions in visual data. Unlike CNNs

that simply use the static convolution kernel to aggregate neighboring features, vision Transformers apply multi-head self-attention to dynamically generate the weights to mix spatial tokens. However, the quadratic complexity w.r.t. the input size of the self-attention largely hinders the application of vision Transformers, especially on downstream tasks including segmentation and detection where higher-resolution feature maps are required. In this work, instead of reducing the complexity of self-attention like previous methods [42, 9, 57], we seek a more efficient and effective way to perform spatial interactions with simple operations like convolution and fully-connected layers.

The basic operation of our method is the gated convolution (gConv). Let $\mathbf{x} \in \mathbb{R}^{HW \times C}$ be the input feature, the output of the gated convolution $\mathbf{y} = g\text{Conv}(\mathbf{x})$ can be written as:

$$
[\mathbf{p}_0^{HW \times C}, \mathbf{q}_0^{HW \times C}] = \phi_{\text{in}}(\mathbf{x}) \in \mathbb{R}^{HW \times 2C},
$$
$$
\mathbf{p}_1 = f(\mathbf{q}_0) \odot \mathbf{p}_0 \in \mathbb{R}^{HW \times C}, \quad \mathbf{y} = \phi_{\text{out}}(\mathbf{p}_1) \in \mathbb{R}^{HW \times C}, \tag{3.1}
$$

where $\phi_{\text{in}}, \phi_{\text{out}}$ are linear projection layers to perform channel mixing and $f$ is a depth-wise convolution. Note that $p_1^{(i,c)} = \sum_{j \in \Omega_i} w_{i \to j}^c q_0^{(j,c)} p_0^{(i,c)}$, where $\Omega_i$ is the local window centered at $i$ and $w$ represents the convolution weight of $f$. Therefore, the above formulation explicitly introduce interactions among the neighboring features $\mathbf{p}_0^{(i)}$ and $\mathbf{q}_0^{(j)}$ through the element-wise multiplication. We consider the interaction in gConv as *1-order interaction* as each $\mathbf{p}_0^{(i)}$ has interacted with its neighbor feature $\mathbf{q}_0^{(j)}$ only once.

**High-order interactions with recursive gating.** After achieving an efficient 1-order spatial interactions with the gConv, we then design the $g^n$Conv, a recursive gated convolution to further enhance the model capacity by introducing higher-order interactions. Formally, we first use $\phi_{\text{in}}$ to obtain a set of projected features $\mathbf{p}_0$ and $\{\mathbf{q}_k\}_{k=0}^{n-1}$:

$$
\left[ \mathbf{p}_0^{HW \times C_0}, \mathbf{q}_0^{HW \times C_0}, \ldots, \mathbf{q}_{n-1}^{HW \times C_{n-1}} \right] = \phi_{\text{in}}(\mathbf{x}) \in \mathbb{R}^{HW \times (C_0 + \sum_{0 \le k \le n-1} C_k)}. \tag{3.2}
$$

We then perform the gated convolution *recursively* by

$$
\mathbf{p}_{k+1} = f_k(\mathbf{q}_k) \odot g_k(\mathbf{p}_k)/\alpha, \qquad k = 0, 1, \ldots, n-1, \tag{3.3}
$$

where we scale the output by $1/\alpha$ to stabilize the training. $\{f_k\}$ are a set of depth-wise convolution layers and $\{g_k\}$ are used to match the dimension in different orders:

$$
g_k = \begin{cases} \text{Identity}, & k = 0, \\ \text{Linear}\,(C_{k-1}, C_k), & 1 \le k \le n-1. \end{cases} \tag{3.4}
$$

Finally, we feed the output of the last recursion step $\mathbf{q}_n$ to the projection layer $\phi_{\text{out}}$ to obtain the result of the $g^n$Conv. From the recursive formula Equation (3.3), it is easy to show that the interaction-order of $\mathbf{p}_k$ will be increased by 1 after each step. As a result, we can see that the $g^n$Conv achieves $n$-order spatial interactions. It is also worth noting that we need only a single $f$ to perform depth-wise convolution to the concatenation of the features $\{\mathbf{q}_k\}_{k=0}^{n-1}$ together instead of computing the convolution in each recursive step as in Equation (3.3), which can further simplify the implementation and improve the efficiency on GPUs. To ensure that the high-order interactions do not introduce too much computational overhead, we set the channel dimension in each order as:

$$
C_k = \frac{C}{2^{n-k-1}}, \qquad 0 \le k \le n-1. \tag{3.5}
$$

This design indicates that we perform the interactions in a coarse-to-fine manner, where lower orders are computed with fewer channels. Besides, the channel dimension of $\phi_{\text{in}}(\mathbf{x})$ is exactly $2C$ and the total FLOPs can be strictly bounded even with $n$ increasing. It can be proved that (see Appendix A):

$$
\text{FLOPs}(g^n\text{Conv}) < HWC(2K^2 + 11/3 \times C + 2), \tag{3.6}
$$

where $K$ is the kernel size of the depth-wise convolution. Therefore, our $g^n$Conv achieves high-order interactions with a similar computational cost to a convolutional layer.

**Long-term interactions with large kernel convolutions.** Another difference between vision Transformers and conventional CNNs is the receptive field. Conventional CNNs [49, 22] often use $3 \times 3$ convolution through the whole network, while vision Transformers calculate self-attention on the whole feature maps [16, 52] or inside a relatively large local window (*e.g.*, $7 \times 7$). The large

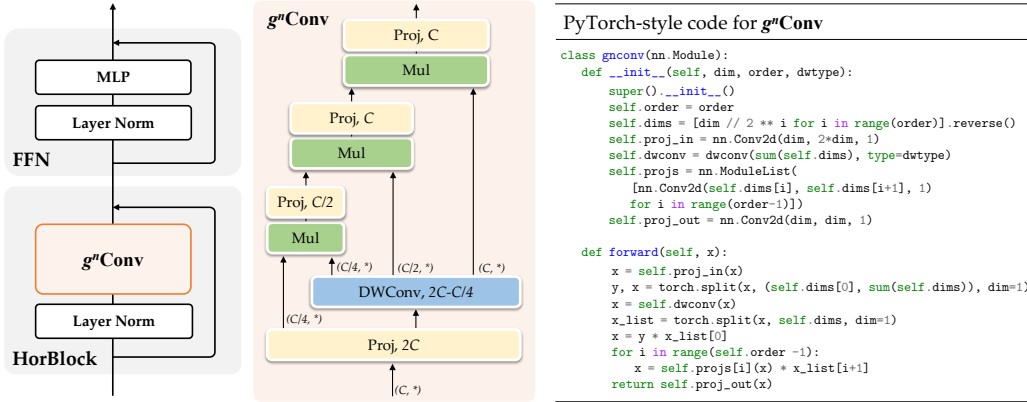

Figure 2: **Overview of the basic building block in HorNet with $g^n$Conv.** We adopt the block design of Transformers [56] and replace the self-attention sub-layer with $g^n$Conv to develop our HorNet (*left*). We also provide the detailed implementation of $g^3$Conv (*middle*) and the Pytorch-style code for an arbitrary order (*right*).

receptive field in vision Transformers makes it easier to capture long-term dependencies, which is also recognized as one of the key advantages of vision Transformers. Inspired by this design, there are some efforts to introduce large kernel convolutions to CNNs recently [14, 43, 46]. To make our $g^n$Conv capable of capturing long-term interactions, we adopt two implementations for the depth-wise convolution $f$:

- *7×7 Convolution*. 7×7 is the default window/kernel size of Swin Transformers [42] and ConvNext [43]. Studies in [43] show that the kernel size produces good performance on ImageNet classification and various downstream tasks. We follow this configuration to fairly compare with representative work of vision Transformers and modern CNNs.

- *Global Filter (GF)*. The GF layer [46] multiplies the frequency domain features with learnable global filters, which is equivalent to a convolution in the spatial domain with a global kernel size and circular padding. We use a modified version of the GF layer by processing half of the channels with the global filter and the other half with 3×3 depth-wise convolutions and only use GF layers in late stages to preserve more local details.

**Spatial interactions in vision models.** We review some representative vision model designs from the perspective of spatial interactions, as shown in Figure 1. Specifically, we are interested in the interactions between a feature $\mathbf{x}_i$ and its neighboring feature $\mathbf{x}_j, j \in \Omega_i$. By using the tool designed for explaining the interaction effect (IE) in [33, 1], we provide an intuitive analysis of the order of explicit spatial interactions in Appendix B. Our analysis reveals a key difference between vision Transformers and previous architectures from a new view, *i.e.*, vision Transformers have higher-order spatial interactions in each basic block. The result inspires us to explore an architecture that can realize more efficient and effective spatial interactions with more than two orders. As discussed above, our proposed $g^n$Conv can achieve arbitrary-order interactions with bounded complexity. It is also worth noting that similar to other scaling factors in deep models like width [69] and depth [22], simply increasing the order of spatial interactions without considering the overall model capacity will not lead to a good trade-off [51]. In this paper, we focus on developing a stronger visual modeling architecture based on the analysis of the spatial interaction orders of well-designed models. We believe a more thorough and formal discussion on the high-order spatial interactions can be an important future direction.

**Relation to dot-product self-attention.** Although the computation of our $g^n$Conv largely differs from dot-product self-attention, we will show that $g^n$Conv also accomplishes the goal of input-adaptive spatial mixing. Let $\mathbf{M}$ be the attention matrix obtained by multi-head self-attention (MHSA), we write $\mathbf{M}$ as $(m_{ij}^c)$ since the mixing weight may vary across the channels. The spatial mixing result (before the final channel mixing projection) of the $c$-th channel at location $i$ is

$$x_{\mathrm{MHSA}}^{(i,c)} = \sum_{j \in \Omega_i} m_{ij}^c v^{(i,j)} = \sum_{j \in \Omega_i} \sum_{c'=1}^{C} \underline{m_{ij}^c} w_V^{(c',c)} x^{(j,c')}, \tag{3.7}$$

where $w_V$ is the weight of the V-projection layer. Note that $m_{ij}$ obtained by the dot-product operation contains 1-order interaction. On the other hand, the output of our $g^n$Conv (before the $\phi_{\text{out}}$) can be written as

$$x_{g^n\text{Conv}}^{(i,c)} = p_n^{(i,c)} = \sum_{j\in\Omega_i}\sum_{c'=1}^{C} \underline{w_{n-1,i\to j}^c \mathbf{g}_{n-1}^{(i,c)}} w_{\phi_{\text{in}}}^{(c',c)} x^{(j,c')} \triangleq \sum_{j\in\Omega_i}\sum_{c'=1}^{C} \underline{h_{ij}^c} w_{\phi_{\text{in}}}^{(c',c)} x^{(j,c')}, \quad (3.8)$$

where $w_{n-1}$ is the convolutional weight for $f_{n-1}$, $w_{\phi_{\text{in}}}$ is the linear weight of $\phi_{\text{in}}$, and $\mathbf{g}_{n-1} = g_{n-1}(\mathbf{p}_{n-1})$ is a projection of $\mathbf{p}_{n-1}$. From the formulation in Equation (3.8) we find our $g^n$Conv also achieves input-adaptive spatial mixing with $\{h_{ij}^c\}$ as the weights. Observing that $h_{ij}$ is computed from $\mathbf{p}_{n-1}$ which contains $n-1$ order interactions, we can regard our $g^n$Conv as an extension of the self-attention in terms of the order of the spatial mixing weight. Therefore, our $g^n$Conv can better model more complex spatial interactions.

The details of $g^n$Conv and our implementation are summarized in Figure 2.

## 3.2 Model Architectures

**HorNet.** The $g^n$Conv can be a drop-in replacement of the spatial mixing layer in vision Transformers [52, 42] or modern CNNs [43]. We follow the same meta-architecture as [56, 42] to construct HorNet, where the basic block contains a spatial mixing layer and a feed-forward network (FFN). Depending on the model size and the implementation of the depth-wise convolution $f_k$ in our $g^n$Conv, we have two series of model variants named HorNet-T/S/B/L$_{7\times7}$ and HorNet-T/S/B/L$_{\text{GF}}$. We consider the popular Swin Transformer [42] and ConvNeXt [43] as the vision Transformer and CNN baselines since our models are implemented based on a convolution-based framework while having high-order interactions like vision Transformers. To fairly compare with the baselines, we directly follow the number of blocks of Swin Transformers-S/B/L [42] but insert an extra block to the stage 2 to make the overall complexity close, resulting in $[2, 3, 18, 2]$ blocks in each stage in all of the model variants. We simply adjust the base number of channels $C$ to construct models with different sizes and set the number of channels in 4 stages as $[C, 2C, 4C, 8C]$ following common practice. We use $C = 64, 96, 128, 192$ for HorNet-T/S/B/L, respectively. We set the interaction orders (*i.e.*, the $n$ in $g^n$Conv) for each stage as 2,3,4,5 by default, such that the channels of the coarsest order $C_0$ is the same across different stages.

**HorFPN.** Apart from using $g^n$Conv in visual encoders, we find our $g^n$Conv can be an enhanced alternative for standard convolution that considers higher-order spatial interactions in a wide range of convolution-based models. Thus, we replace spatial convolutions for feature fusion in the FPN [37] with our $g^n$Conv to improve spatial interactions for downstream tasks. Specifically, we add our $g^n$Conv after the fusion of features from different pyramid levels. For object detection, we replace the 3×3 convolution after the top-down pathway with the $g^n$Conv in each level. For semantic segmentation, we simply replace the 3×3 convolution after the concatenation of the multi-level feature maps with $g^n$Conv since the final results are directly predicted from this concatenated feature. We also have two implementations called HorFPN$_{7\times7}$ and HorFPN$_{\text{GF}}$ decided by the choice of $f_k$.

# 4 Experiments

We conduct extensive experiments to verify the effectiveness of our method. We present the main results on ImageNet [13] and compare them with various architectures. We also test our models on the downstream dense prediction tasks on commonly used semantic segmentation benchmark ADE20K [71] and object detection dataset COCO [38]. Lastly, we provide ablation studies of our designs and analyze the effectiveness of $g^n$Conv on a wide range of models.

## 4.1 ImageNet Classification

**Setups.** We conduct image classification experiments on the widely used ImageNet [13] dataset. We train our HorNet-T/S/B models using the standard ImageNet-1K dataset following common practice. To fairly compare with previous work, we directly use the training configurations of [43, 42, 52] to train our models. We train the models for 300 epochs with $224 \times 224$ input. To evaluate the scaling ability of our designs, we further train the HorNet-L models on the ImageNet-22K dataset that contains over $10\times$ images and more categories. We follow previous practice [42, 43] to train our models for 90 epochs and use a similar data augmentation strategy as ImageNet-1K experiments.

Table 1: **ImageNet classification results.** We compare our models with state-of-the-art vision Transformers and CNNs that have comparable FLOPs and parameters. We report the top-1 accuracy on the validation set of ImageNet as well as the number of parameters and FLOPs. We also show the improvements over Swin Trasnformers that have similar overall architectures and training configurations to our models. "↑384" indicates that the model is fine-tuned on 384×384 images for 30 epochs. Our models are highlighted in gray.

| Model | Image Size | Params (M) | FLOPs (G) | Top-1 Acc. (%) | Model | Image Size | Params (M) | FLOPs (G) | Top-1 Acc. (%) |
|---|---|---|---|---|---|---|---|---|---|
| *ImageNet-1K trained models* | | | | | *ImageNet-1K trained models (fine-tuned at 384×384)* | | | | |
| EfficientNet-B4 [51] | $380^2$ | 19 | 4.2 | 82.9 | Swin-B↑384 [42] | $384^2$ | 89 | 47.1 | 84.5 |
| EfficientNet-B5 [51] | $456^2$ | 30 | 9.9 | 83.6 | ConvNeXt-B↑384 [43] | $384^2$ | 88 | 45.0 | $85.1_{(+0.6)}$ |
| EfficientNet-B6 [51] | $528^2$ | 43 | 19.0 | 84.0 | HorNet-B$_{7\times7}$↑384 | $384^2$ | 87 | 45.8 | $85.3_{(+0.8)}$ |
| EfficientNetV2-S [51] | $300^2$ | 24 | 8.8 | 83.9 | HorNet-B$_{GF}$↑384 | $384^2$ | 92 | 45.4 | $\mathbf{85.6}_{(+1.1)}$ |
| RepLKNet-31B [14] | $224^2$ | 79 | 15.3 | 83.5 | | | | | |
| VAN-B [19] | $224^2$ | 27 | 5.0 | 82.8 | *ImageNet-22K trained models (fine-tuned to ImageNet-1K)* | | | | |
| VAN-L [19] | $224^2$ | 45 | 9.0 | 83.9 | R-101x3 [30] | $384^2$ | 388 | 204.6 | 84.4 |
| CSWin-T [15] | $224^2$ | 23 | 4.3 | 82.7 | R-152x4 [30] | $480^2$ | 937 | 840.5 | 85.4 |
| CSWin-S [15] | $224^2$ | 35 | 6.9 | 83.6 | ViT-B/16 [16] | $384^2$ | 87 | 55.5 | 84.0 |
| CSWin-B [15] | $224^2$ | 78 | 15.0 | 84.2 | ViT-L/16 [16] | $384^2$ | 305 | 191.1 | 85.2 |
| Swin-T [42] | $224^2$ | 28 | 4.5 | 81.3 | EfficientNetV2-L [51] | $380^2$ | 121 | 53.0 | 86.8 |
| ConvNeXt-T [43] | $224^2$ | 29 | 4.5 | $82.1_{(+0.7)}$ | CSWin-L [15] | $384^2$ | 173 | 96.8 | 87.5 |
| HorNet-T$_{7\times7}$ | $224^2$ | 22 | 4.0 | $82.8_{(+1.5)}$ | SwinV2-L [41] | $384^2$ | 197 | 115.4 | 87.6 |
| HorNet-T$_{GF}$ | $224^2$ | 23 | 3.9 | $\mathbf{83.0}_{(+1.7)}$ | RepLKNet-31L [14] | $384^2$ | 172 | 96.0 | 86.6 |
| Swin-S [42] | $224^2$ | 50 | 8.7 | 83.0 | Swin-L [42] | $224^2$ | 197 | 34.5 | 86.3 |
| ConvNeXt-S [43] | $224^2$ | 50 | 8.7 | $83.1_{(+0.1)}$ | ConvNeXt-L [43] | $224^2$ | 198 | 34.4 | $86.6_{(+0.3)}$ |
| HorNet-S$_{7\times7}$ | $224^2$ | 50 | 8.8 | $83.8_{(+0.8)}$ | HorNet-L$_{7\times7}$ | $224^2$ | 195 | 34.8 | $86.8_{(+0.5)}$ |
| HorNet-S$_{GF}$ | $224^2$ | 50 | 8.7 | $\mathbf{84.0}_{(+1.0)}$ | HorNet-L$_{GF}$ | $224^2$ | 196 | 34.6 | $\mathbf{87.0}_{(+0.7)}$ |
| Swin-B [42] | $224^2$ | 89 | 15.4 | 83.5 | Swin-L↑384 [42] | $384^2$ | 197 | 103.9 | 87.3 |
| ConvNeXt-B [43] | $224^2$ | 88 | 15.4 | $83.8_{(+0.3)}$ | ConvNeXt-L↑384 [43] | $384^2$ | 198 | 101.0 | $87.5_{(+0.2)}$ |
| HorNet-B$_{7\times7}$ | $224^2$ | 87 | 15.6 | $84.2_{(+0.7)}$ | HorNet-L$_{7\times7}$↑384 | $384^2$ | 195 | 102.3 | $87.6_{(+0.3)}$ |
| HorNet-B$_{GF}$ | $224^2$ | 88 | 15.5 | $\mathbf{84.3}_{(+0.8)}$ | HorNet-L$_{GF}$↑384 | $384^2$ | 202 | 101.8 | $\mathbf{87.7}_{(+0.4)}$ |

We fine-tune the models pre-trained on ImageNet-22K or at the 224×224 resolution to ImageNet-1K or/and 384×384 resolution for 30 epochs following [43]. When adapting the ImageNet-22K models to ImageNet-1K, we initialize the classifier with the pre-trained class centers to stabilize the training process. More details can be found in Appendix C.

**Results.** The results of our ImageNet classification experiments are summarized in Table 1. We see that our models achieve very competitive performance with state-of-the-art vision Transformers and CNNs. Notably, HorNet surpasses Swin Transformers and ConvNeXt which have similar overall architectures and training configurations by a healthy margin on various model sizes and settings. Our models also generalize well to a larger image resolution, larger model sizes and more training data. These results clearly demonstrate the effectiveness and generality of our designs.

## 4.2 Dense Prediction Tasks

**HorNet for semantic segmentation.** We evaluate our HorNet for semantic segmentation task on ADE20K [71] dataset using the commonly used UperNet [62] framework. All the models are trained for 160k iterations using AdamW [44] optimizer with a global batch size of 16. The image size during training is $512 \times 512$ for ImagNet-1k (HorNet-T/S/B) pre-trained models and $640 \times 640$ for the ImageNet-22K pre-trained models (HorNet-L). The results are summarized in the left part of Table 2, where we report both the single-scale (SS) and multi-scale (MS) mIoU on the validation set. Both our HorNet$_{7\times7}$ and HorNet$_{GF}$ models outperform Swin [42] and ConvNeXt [43] models with similar model sizes and FLOPs. Specifically, HorNet$_{GF}$ models achieve better results than HorNet$_{7\times7}$ and ConvNeXt series by large margins in single-scale mIoU, indicating the global interactions captured by the global filter are helpful for semantic segmentation. Notably, we find both our HorNet-L$_{7\times7}$ and HorNet-L$_{GF}$ even outperform ConvNeXt-XL with ∼25% fewer FLOPs. These results clearly demonstrate the effectiveness and scalability of our HorNet on semantic segmentation.

**HorNet for object detection.** We also evaluate our models on the COCO [38] dataset. We adopt the cascade Mask R-CNN framework [21, 2] to perform object detection and instance segmentation using HorNet-T/S/B/L backbones. Following Swin [42] and ConvNeXt [43], we use $3\times$ schedule with multi-scale training. The right part of Table 2 compares the box AP and mask AP of our HorNet models and Swin/ConvNeXt models. Similarly, we show our HorNet models achieve consistently and significantly better performance than the Swin/ConvNeXt counterparts, in both box AP and mask AP. The HorNet$_{GF}$ series obtain +1.2∼2.0 box AP and +1.0∼1.9 mask AP compared with ConvNeXt.

Table 2: **Object detection and semantic segmentation results with different backbones.** We use UperNet [62] for semantic segmentation and Cascade Mask R-CNN [2] for object detection. ‡ indicates that the model is pre-trained on ImageNet-22K. For semantic segmentation, we report both single-scale (SS) and multi-scale (MS) mIoU. The FLOPs are calculated with image size (2048, 512) for ImageNet-1K pre-trained models and (2560, 640) for ImageNet-22K pre-trained models. For object detection, we report the box AP and the mask AP. FLOPs are measured on input sizes of (1280, 800). Our models are highlighted in gray.

| Backbone | Semantic Segmentation with *UperNet 160K* | | | | Object Detection with *Cascade Mask R-CNN 3×* | | | |
|---|---|---|---|---|---|---|---|---|
| | $mIoU^{ss}$ | $mIoU^{ms}$ | Params | FLOPs | $AP^{box}$ | $AP^{mask}$ | Params | FLOPs |
| Swin-T [42] | 44.5 | 45.8 | 60M | 945G | 50.4 | 43.7 | 86M | 745G |
| ConvNeXt-T [43] | 46.0 | 46.7 | 60M | 939G | 50.4 | 43.7 | 86M | 741G |
| HorNet-T$_{7\times7}$ | 48.1 | 48.9 | 52M | 926G | 51.7 | 44.8 | 80M | 730G |
| HorNet-T$_{GF}$ | **49.2** | **49.3** | 55M | 924G | **52.4** | **45.6** | 80M | 728G |
| Swin-S [42] | 47.6 | 49.5 | 81M | 1038G | 51.9 | 45.0 | 107M | 838G |
| ConvNeXt-S [43] | 48.7 | 49.6 | 82M | 1027G | 51.9 | 45.0 | 108M | 827G |
| HorNet-S$_{7\times7}$ | 49.2 | 49.8 | 81M | 1030G | 52.7 | 45.6 | 107M | 830G |
| HorNet-S$_{GF}$ | **50.0** | **50.5** | 85M | 1027G | **53.3** | **46.3** | 108M | 827G |
| Swin-B [42] | 48.1 | 49.7 | 121M | 1188G | 51.9 | 45.0 | 145M | 982G |
| ConvNeXt-B [43] | 49.1 | 49.9 | 122M | 1170G | 52.7 | 45.6 | 146M | 964G |
| HorNet-B$_{7\times7}$ | 50.0 | 50.5 | 121M | 1174G | 53.3 | 46.1 | 144M | 969G |
| HorNet-B$_{GF}$ | **50.5** | **50.9** | 126M | 1171G | **54.0** | **46.9** | 146M | 965G |
| Swin-L$^{\ddagger}$ [42] | 52.1 | 53.5 | 234M | 2468G | 53.9 | 46.7 | 253M | 1382G |
| ConvNeXt-L$^{\ddagger}$ [43] | 53.2 | 53.7 | 235M | 2458G | 54.8 | 47.6 | 255M | 1354G |
| ConvNeXt-XL$^{\ddagger}$ [43] | 53.6 | 54.0 | 391M | 3335G | 55.2 | 47.7 | 407M | 1898G |
| HorNet-L$^{\ddagger}_{7\times7}$ | 54.1 | 54.5 | 232M | 2473G | 55.4 | 48.0 | 251M | 1363G |
| HorNet-L$^{\ddagger}_{GF}$ | **55.0** | **55.2** | 239M | 2465G | **56.0** | **48.6** | 259M | 1358G |

Table 3: **Comparisons of HorFPN with standard FPN on different backbones.** We use UperNet 160K and Mask R-CNN 1× schedule for semantic segmentation and object detection, respectively. We find our HorFPN consistently outperforms standard FPN with various of backbones on both the two tasks.

| Backbone | Fusion Module | Semantic Segmentation with *UperNet 160K* | | | | Object Detection with *Mask R-CNN 1×* | | | |
|---|---|---|---|---|---|---|---|---|---|
| | | $mIoU^{ss}$ | $mIoU^{ms}$ | Params | FLOPs | $AP^{box}$ | $AP^{mask}$ | Params | FLOPs |
| ResNet-50 [22] | FPN [37] | 40.7 | 41.8 | 66M | 947G | 38.2 | 34.7 | 44M | 260G |
| | HorFPN$_{7\times7}$ | 41.8 | 44.1 | 60M | 499G | 38.7 | 35.1 | 43M | 226G |
| | HorFPN$_{GF}$ | **43.2** | **44.5** | 60M | 497G | **39.1** | **35.5** | 43M | 224G |
| ResNet-101 [22] | FPN [37] | 42.9 | 44.0 | 85M | 1025G | 40.0 | 36.1 | 63M | 336G |
| | HorFPN$_{7\times7}$ | 44.1 | 45.5 | 79M | 577G | 40.3 | 36.4 | 62M | 302G |
| | HorFPN$_{GF}$ | **44.5** | **46.4** | 79M | 574G | **40.5** | **36.7** | 62M | 300G |
| Swin-S [42] | FPN [37] | 47.6 | 49.5 | 81M | 1038G | 45.5 | 40.9 | 69M | 354G |
| | HorFPN$_{7\times7}$ | 48.0 | 49.2 | 74M | 580G | 46.3 | 41.1 | 68M | 325G |
| | HorFPN$_{GF}$ | **49.0** | **49.9** | 75M | 578G | **46.8** | **41.9** | 69M | 323G |
| HorNet-S | FPN [37] | 49.2 | 49.8 | 81M | 1030G | 47.1 | 42.2 | 69M | 351G |
| | HorFPN$_{7\times7}$ | 49.4 | 50.1 | 74M | 577G | 47.4 | 42.3 | 68M | 322G |
| | HorFPN$_{GF}$ | **49.7** | **50.3** | 75M | 575G | **47.7** | **42.4** | 68M | 321G |

Again, our large model HorNet-L$_{7\times7}$ and HorNet$_{GF}$ can outperform ConvNeXt-XL, which further validates the favorable transferability with a larger model size and larger pre-trained dataset.

**HorFPN for dense prediction.** We now show another application of the proposed $g^nConv$, *i.e.*, to serve as a better fusion module that can better capture the higher-order interactions among different levels of features in dense prediction tasks. Specifically, we directly modify the FPN [37] as described in Section 3.2 in UperNet [62] and Mask R-CNN [21] for semantic segmentation and object detection, respectively. We show the results in Table 3, where we compare the performance of our HorFPN and standard FPN on different backbones including ResNet-50/101 [22], Swin-S [42] and HorNet-S$_{7\times7}$. For semantic segmentation, we find our HorFPN can significantly reduce the FLOPs ($\sim$50%) while achieving better validation mIoU. For object detection,

Table 4: **Object detection results with recent state-of-the-art frameworks.** We report the single-scale $AP^{box}$ and $AP^{mask}$ on the validation set of COCO. Our models are highlighted in gray.

| Backbone | Framework | $AP^{box}$ | $AP^{mask}$ |
|---|---|---|---|
| Swin-L [42] | HTC++ [3] | 57.1 | 49.5 |
| ViT-Adapter-L [5] | HTC++ [3] | 57.9 | 50.2 |
| HorNet-L$_{GF}$ | HTC++ [3] | **58.1** | **50.5** |
| Swin-L [42] | DINO [70] | 58.5 | - |
| HorNet-L$_{GF}$ | DINO [70] | **59.2** | - |

Table 5: **Semantic Segmentation results with recent state-of-the-art frameworks.** We report the single-scale (SS) and multi-scale (MS) mIoU on the validation set of ADE20K. Our models are highlighted in gray.

| Backbone | Framework | $mIoU^{ss}$ | $mIoU^{ms}$ |
|---|---|---|---|
| Swin-L [42] | Mask2Former [7] | 56.1 | 57.3 |
| Swin-L-FaPN [27] | Mask2Former [7] | 56.4 | 57.7 |
| HorNet-L$_{GF}$ | Mask2Former [7] | **57.5** | **57.9** |

Table 6: **Ablation study and results of applying $g^n$Conv to other models/operations.** We provide the ablation study of our designs in (a). [*] indicates the baseline of our model. The baseline and our final models are highlighted in gray. In (b) and (c), we apply the proposed $g^n$Conv to isotropic models that have a similar level of complexity with ViT/DeiT-S [16, 52] and other spatial mixing operations including the 3×3 depth-wise convolution and 3×3 pooling used in [67].

(a) Ablation study.

| Model | Params | FLOPs | Acc. (%) |
|---|---|---|---|
| Swin-T [42] | 28M | 4.5G | $81.3_{(+0.1)}$ |
| - Self-Attention + DWConv$_{7\times7}$ [*] | 29M | 4.5G | 81.2 |
| + SE [25] | 30M | 4.5G | $81.5_{(+0.3)}$ |
| - SE + $g^{\{1,1,1,1\}}$Conv | 28M | 4.3G | $81.7_{(+0.5)}$ |
| + $g^{\{2,2,2,2\}}$Conv | 28M | 4.3G | $82.2_{(+1.0)}$ |
| + $g^{\{3,3,3,3\}}$Conv | 28M | 4.3G | $82.5_{(+1.3)}$ |
| + $g^{\{4,4,4,4\}}$Conv | 28M | 4.3G | $82.5_{(+1.3)}$ |
| + $g^{\{1,2,3,4\}}$Conv | 28M | 4.3G | $82.5_{(+1.3)}$ |
| + $g^{\{2,3,4,5\}}$Conv | 28M | 4.3G | $82.6_{(+1.4)}$ |
| + Deeper & Narrower (HorNet-T$_{7\times7}$) | 22M | 4.0G | $82.8_{(+1.6)}$ |
| + Global Filters [46] (HorNet-T$_{GF}$) | 23M | 3.9G | $83.0_{(+1.8)}$ |
| ConvNeXt [43] | 28M | 4.5G | $82.1_{(+0.9)}$ |

(b) Results on isotropic models.

| Model | FLOPs | Acc. (%) |
|---|---|---|
| DeiT-S [52] | 4.6G | 79.8 |
| ConvNeXt-S (iso.) [43] | 4.3G | 79.7 |
| HorNet-S$_{7\times7}$ (iso.) | 4.5G | 80.6 |
| HorNet-S$_{GF}$ (iso.) | 4.5G | 81.0 |

(c) $g^n$Conv for other operations.

| Model | FLOPs | Acc. (%) |
|---|---|---|
| DWConv$_{3\times3}$ | 4.0G | 80.7 |
| $g^n$Conv$_{3\times3}$ | 3.9G | 82.1 |
| Pool [67] | 3.9G | 78.1 |
| $g^n$Conv$_{pool}$ | 3.8G | 79.3 |

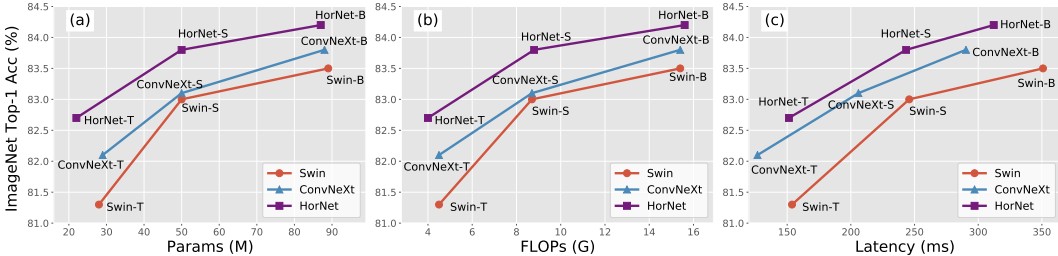

Figure 3: **Comparisons of trade-offs of Swin, ConvNeXt and HorNet.** We compare the trade-offs of the models via the top-1 accuracy on ImageNet w.r.t. **(a)** number of parameters; **(b)** FLOPs; **(c)** latency. The latency is measured with a single NVIDIA RTX 3090 GPU with a batch size of 128.

our HorFPN can also outperform standard FPN in terms of both box AP and mask AP on different backbones with about 30G fewer FLOPs. Besides, we observe that the HorFPN$_{GF}$ is consistently better than HorFPN$_{7\times7}$, indicating that global interactions are also important when fusing hierarchical features.

**Results with state-of-the-art frameworks.** To further show the effectiveness our backbone, we conduct experiments to combine our large HorNet model with recent state-of-the-art dense prediction frameworks including HTC++ [3], DINO [70] and Mask2Former [7]. For HTC++ and DINO, we train our models on COCO for 36 epochs (3× schedule) and does not introduce extra pre-training data like Object365 in [70]. We report the single-scale performance on the validation set and compared with several state-of-the-art methods in Table 4. For Mask2Former, we train our models on ADE20K with $640 \times 640$. We report the mIoU of both single-scale and multi-scale testing on the validation set in Table 5.

### 4.3 Analysis

**Ablation study.** We provide detailed ablation studies of the $g^n$Conv and our HorNet in Table 6. We first study the model designs of our HorNet in Table 6a. Our baseline ([*]) is obtained by simply replacing the self-attention with 7×7 depth-wise convolution in Swin-T [42]. We first show that both SE [25] and our $g^n$Conv with $n = 1$ ($g^{\{1,1,1,1\}}$Conv) can improve over the baseline model [*], and $g^{\{1,1,1,1\}}$Conv is slightly better. We then perform ablations on the interaction order $n$ for each stage and find: (1) if $n$ is shared across the 4 stages, the accuracy will increase with larger $n$ but saturate at 82.5 when $n = 4$; (2) progressively increased order ($g^{\{2,3,4,5\}}$Conv) can further improve the accuracy. Our final models are built on $g^{\{2,3,4,5\}}$Conv by adjusting the depth and width of the networks (HorNet-T$_{7\times7}$) and applying Global Filter [46] for the depth-wise convolution (HorNet-T$_{GF}$). These results clearly show that our $g^n$Conv is an efficient and extendable operation that can better capture high-order spatial interactions than both self-attention and depth-wise convolution.

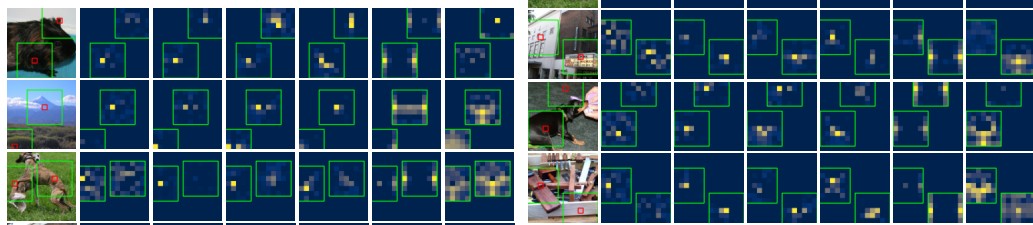

Figure 4: Visualization of the adaptive weights generated by $g^n$Conv. We see that the spatial mixing weights of our $g^n$Conv are adaptive both to input samples and spatial locations, which further indicates that $g^n$Conv shares these two desirable characteristics with the self-attention operation.

**$g^n$Conv for isotropic models.** We also evaluate $g^n$Conv on isotropic architectures (with constant spatial resolutions). We replace the self-attention in DeiT-S [52] with our $g^n$Conv and adjust the number of blocks to 13 to obtain the isotropic HorNet-S$_{7\times7}$ and HorNet-S$_{\text{GF}}$. We compare DeiT-S, isotropic ConvNeXt-S and isotropic HorNet-S in Table 6b. While isotropic ConvNeXt-S cannot improve DeiT-S, our isotropic HorNet surpasses DeiT-S by a large margin. These results indicate that our $g^n$Conv can better realize the functions of self-attention compared to plain convolutions and have better ability to model the complex spatial interactions.

**$g^n$Conv for other operations.** To further demonstrate the universality of $g^n$Conv, we use $3\times3$ depth-wise convolution and $3\times3$ pooling [67] as the basic operation in the $g^n$Conv. The results in Table 6c show that $g^n$Conv can also improve these two operations by large margins, indicating our $g^n$Conv is potentially more powerful when equipped with some better basic operations.

**Accuracy-complexity trade-offs.** We visualize accuracy-complexity trade-offs of Swin, ConvNeXt and HorNet series in Figure 3. For fair comparisons, we fix the input image size to $224 \times 224$ and use HorNet$_{7\times7}$ such that all the compared models are based on $7\times7$ local window. We see HorNet can achieve better trade-offs than the representative vision Transformers and modern CNNs with regards to model size, FLOPs and GPU latency.

**Visualization.** We provide some visualizations of the adaptive weights learned by $g^n$Conv in Figure 4. For each sample, we show the value of $\frac{1}{C}\sum_{c=1}^{C} h_{ij}^c$ (see Equation (3.8) or the definition of $h_{ij}^c$) for two random spatial locations $i$ from layer $\{1, 3, 5, 7, 8, 12\}$ of the isotropic HorNet-S model. Figure 4 demonstrates that the spatial mixing weights of our $g^n$Conv are adaptive both to input samples and spatial locations, which further indicates that $g^n$Conv shares these two desirable characteristics with the self-attention operation.

**Limitations.** While HorNet shows better overall latency-accuracy trade-offs, we notice that HorNet is slower than ConvNeXt with similar FLOPs on GPU, which may be caused by the more complex designs to perform the high-order interactions. We think that developing a more hardware-friendly operation for high-order spatial interactions is an interesting future direction to improve our work.

## 5   Conclusion

We have presented the Recursive Gated Convolution ($g^n$Conv) that performs efficient, extendable, and translation-equivariant high-order spatial interactions with gated convolutions and recursive deigns. $g^n$Conv can serve as a drop-in replace of the spatial mixing layer in various vision Transformers and convolution-based models. Based on the operation, we have constructed a new family of generic vision backbones HorNet. Extensive experiments demonstrate the effectiveness of $g^n$Conv and HorNet on commonly used visual recognition benchmarks. We hope our attempt can inspire future work to further explore the high-order spatial interactions in vision models.

## Acknowledgments

Jiwen Lu was supported in part by the National Key Research and Development Program of China under Grant 2017YFA0700802, the National Natural Science Foundation of China under Grant 62125603 and Grant U1813218, and a grant from the Beijing Academy of Artificial Intelligence (BAAI).

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
