# OpenReview forum: "HorNet: Efficient High-Order Spatial Interactions with Recursive Gated Convolutions"
_NeurIPS.cc/2022/Conference — NeurIPS 2022 Accept_

### Official Review · Reviewer_3Snz · 2022-07-01

**Rating:** 8
**Confidence:** 5
**Soundness:** 4 excellent
**Presentation:** 4 excellent
**Contribution:** 4 excellent

**Summary:**

This paper proposes a new operation (gnConv) and network (HorNet) to perform computer vision tasks. Motivated by the success of vision Transformers, the key idea is to build an architecture that has input-adaptive, long-range and high-order spatial interactions. The authors did so by proposing a recursive form of gated convolution, which has the capability of going even higher order explicit spatial interaction than 2. On well-established computer vision benchmarks such as image classification on ImageNet, semantic segmentation on ADE20K, object detection on COCO, the authors showed consistent improvement over Swin Transformer and ConvNeXt.

POST-REBUTTAL UPDATE:

I have read the authors' rebuttal. While I am a bit disappointed by the answer to my Q2, I am satisfied with the rebuttal overall and the paper. In my opinion / experience, in this day and age, it is not an easy feat to achieve the consistent improvement as the authors have shown in this paper, and I have decided to maintain my original rating of 8.

**Questions:**

L316 states that HorNet currently runs slower than ConvNeXt. It would be good to clarify how many times slower.

HorNet seems to outperform Swin Transformer and ConvNeXt pretty consistently. But have the authors considered comparison against models with better performance on ImageNet, such as CoAtNet [8]? Related, what are ways that will allow HorNet to achieve state-of-the-art on ImageNet classification?

Finally, I think the higher-order interaction really originated from the gating mechanisms in LSTM. In vision, "SORT: Second-Order Response Transform for Visual Recognition" from ICCV 2017 is one of the earliest works. It would be great to add proper discussion on these.

**Limitations:**

I am reasonably happy with the Limitations section of the paper from L315 to L318. As I wrote earlier, it would be good to be more transparent and specific.

**Strengths And Weaknesses:**

Strengths:
- The proposed architecture is very simple and effective. Higher-order interactions have been desired but have never been prevalent, possibly due to gradient problems. But the authors seem to have got it to work quite well.
- Writing is very clear.
- Experiments are thorough, and solid improvements are observed throughout.

Weaknesses:
- No major weaknesses. I just have a couple of questions.

Overall, I think the originality, quality, clarity, and significance are all high.

---

> ### Author Response · Authors · 2022-08-02
> **Response to Reviewer 3Snz**
>
> We sincerely thank the reviewer for the positive comments on our work! We address the questions and clarify the issues accordingly as described below.
>
> >**Q1: About the throughput of HorNet**
>
> **[Reply]** Thanks for your suggestion. Here we provide the comparisons of the throughput of HorNet and ConvNeXt, along with the FLOPs and the top-1 accuracy on ImageNet.
>
> |Model | GFLOPs| Throughput (images / s) | Acc. (%)|
> |------|-------|--------|-------|
> |ConvNeXt-T| 4.5 |   1010.3   |82.1 |
> |HorNet-T$_{7\times 7}$| 4.0 |   845.7   |82.7 |
> |ConvNeXt-S| 8.7 |   621.5   |83.1 |
> |HorNet-S$_{7\times 7}$| 8.8 |   525.8   |83.8 |
> |ConvNeXt-B| 15.4 |  440.8   |83.8 |
> |HorNet-B$_{7\times 7}$| 15.6 |   410.0   |84.2 |
>
> As can be seen in the above table, HorNet runs about 15% slower than ConvNeXt for tiny and small models. For base models, the gap is reduced to about 7%. We admit that at current time we have not found a more hardware-friendly implementation of the $\textit{g}^\textit{n}\text{Conv}$. However, we note that our HorNet series still enjoy better speed-accuracy trade-offs than ConvNeXt series, as shown in Fig. 3(c). We think that with more careful hardware-friendly designs, our $\textit{g}^\textit{n}\text{Conv}$ can serve as a general and efficient operation that can be used as a drop-in replacement of self-attention to boost the performance of various vision backbones.
>
> >**Q2: About the comparisons with better performance on ImageNet.**
>
> **[Reply]** Thanks for your insightful question. The main goal of our experiments is to verify the effectiveness of our new operation. Therefore, we focus on comparing our models with typical ConvNets and vision Transformers using similar architectures and training configurations. Due to the limit of computational resources, we cannot conduct very large-scale experiments like CoAtNet (billions of parameters models or training dataset larger than ImageNet-22K). Except for scaling up the model size and training data, we think the performance of our models can be further improved from the following perspectives: 1) more optimized overall architectures (optimized depth/width for each stage), 2) better patch embedding strategies (overlapping convolutional layers for input embedding and downsampling), 3) more advanced training methods, 4) more efficient ways to produce adaptive weights (using downsmapled features to produce attention weights like MViT),  5) hybrid architectures (combining $\textit{g}^\textit{n}\text{Conv}$ with self-attention and plain convolutions). We have added some discussions on how to further improve our models in the revised appendix (supplementary material). Besides, as mentioned in our reply to Reviewer j8qU, we also show that HorNet can achieve state-of-the-art level performance on downstream object detection and semantic segmentation in the setting of not using further inference tricks or extra pre-training data.
>
> >**Q3: About more discussions on previous works.**
>
> **[Reply]** Thanks for pointing out this. We notice that the mentioned paper SORT also uses element-wise multiplication to introduce second-order interactions, and our $\textit{g}^\textit{n}\text{Conv}$ shares the similar idea. The main difference between $\textit{g}^\textit{n}\text{Conv}$ and SORT is that $\textit{g}^\textit{n}\text{Conv}$ is more extendable to achieve higher order spatial interactions under a controllable computational budget. We have included more discussions about the previous works on the higher-order interaction, as well as the gating mechanisms in LSTM in our revised version.

---

> > ### Author Response · Authors · 2022-08-07
> > **Looking forward to your feedback**
> >
> > Dear reviewer 3Snz,
> >
> > Does our response address your concerns? Please feel free to let us know if you have any further questions.
> >
> > Best wishes!

---

### Official Review · Reviewer_j8qU · 2022-07-04

**Rating:** 5
**Confidence:** 5
**Soundness:** 3 good
**Presentation:** 3 good
**Contribution:** 3 good

**Summary:**

This paper proposes a new convolution-based deep neural architecture named $g^nConv$. Its essential operation is the gated convolution which multiplies the output of linear projection layers and depth-wise convolution layers. It also introduces high-order interactions with recursive gating. The authors conduct various experiments to show the effectiveness of the proposed results, including ImageNet classification, COCO object detection, and ADE20K semantic segmentation.

**Questions:**

1. It would be better if the authors could theoretically analyze why High-order interactions can improve network performance.
2. The authors are suggested to compare with more recent works and evaluate the throughput of each method.

**Limitations:**

Yes

**Strengths And Weaknesses:**

Strengths:
1. This is a well-written paper with carefully designed experiments.
2. This paper combines several deep learning techniques in an interesting way to get a novel architecture for image classification, detection, and segmentation.

Weaknesses:
1. The experimental result section is weak. The authors did not compare their proposed method with the state-of-the-art methods. The authors are suggested to add more recent works for comparison, e.g., Dynamic Group Transformer [Liu etal, IJCAI 2022], MViTv2 [Li etal, CVPR 2022], Pale Transformer [Wu etal, AAAI 2022]
2. The authors did not compare the throughput of each method. Since the proposed method involves a lot of small matrix multiplications that cannot be parallelized, and depth-wise convolutions, I guess this method should be slower than the other methods, even though their FLOPs are similar.

---

> ### Author Response · Authors · 2022-08-02
> **Response to Reviewer j8qU, Part I**
>
> We sincerely thank the reviewer for the detailed comments and insightful advice. We address the questions and clarify the issues accordingly as described below.
>
> >**Q1: About the experimental results and the comparisons with recent methods**
>
> **[Reply]** Thanks for your detailed suggestions. We agree that our models cannot outperform some recent methods like Dynamic Group Transformer [r1] and Pale Transformer [r3]. However, it is worth noting that the goal of our experiments is not to achieve state-of-the-art performance on ImageNet-1K, but to demonstrate the effectiveness of the new basic operation and the new family of architectures (HorNet). We also didn’t claim our models can state-of-the-art performance on ImageNet.
>
> - **i. Our main contribution is the new basic operation instead of a visual recognition system that can achieve state-of-the-art performance.** Our experiments are designed to clearly verify the superior of our design over previous basic operations like plain convolution and self-attention. Therefore, we choose to strictly follow the basic architecture and the training configuration of widely used architectures Swin Transformers and ConvNeXt (See our descriptions in Section 3.2). Our goal is to provide a new and useful basic operation for future research, instead of developing a state-of-the-art visual recognition system. We believe both directions are very important for deep learning and the computer vision community. Note that many very impactful methods like Swin Transformers and ConvNeXt also didn’t achieve the best performance on ImageNet-1K.
>
> - **ii. Our performance on ImageNet-1K can be further improved if more advanced and complex designs are adopted.** As mentioned above, we directly adopt the widely used architectures in our experiments. Therefore, there is still substantial room to further improve the performance on ImageNet-1K. We think many techniques that have been used in previous work can be useful, including further optimized overall architectures (optimized depth/width for each stage), better patch embedding strategies (overlapping convolutional layers for input embedding and downsampling), more efficient ways to produce adaptive weights (using downsampled features to produce attention weights like MViT), more advanced training methods and hybrid architectures (combining $\textit{g}^\textit{n}\text{Conv}$ with self-attention and plain convolutions).
>
> - **iii. Our simple architectures based on Swin Transformers are easy to combine with state-of-the-art frameworks on downstream tasks.** Since most recent state-of-the-art object detection and semantic segmentation frameworks are designed and tuned on Swin Transformers, we can directly apply these methods to HorNet without further tuning. To further show the potential of our model on downstream tasks, we apply our HorNet-L model to recent high-performance dense prediction frameworks including HTC++ [A], DINO [B], and Mask2Former [C]. The results are listed in the following table. We see our method can achieve state-of-the-art level performance on COCO and ADE20K in the setting of not using further inference tricks (e.g., TTA for object detection) or extra pre-training data (e.g., Object365 [D] pre-training for COCO or COCO-stuff [E] pre-training for ADE20K).
>
> Object detection results:
> |Model| Framework| mAP$^{box}$ |  mAP$^{mask}$ |
> |------|-------|--------|-------|
> |Swin-L | HTC++|  57.1 | 49.5|
> |ViT-Adapter-L [F] | HTC++ | 57.9 | 50.2 |
> | HorNet-L | HTC++ | **58.1** | **50.5** |
> |Swin-L | DINO |  57.6 |  - |
> |HorNet-L | DINO |  **59.2** | - |
>
> Semantic segmentation results:
> |Model| Framework| mIoU$^{ss}$ |  mIoU$^{ms}$ |
> |------|-------|--------|-------|
> |Swin-L | Mask2Former | 56.1 |  57.3 |
> |Swin-L | Mask2Former |**57.5** |  **57.9**|
>
> We will include discussions about these suggested papers and include some discussions about how to achieve better performance on ImageNet-1K with $\textit{g}^\textit{n}\text{Conv}$ in the revised paper. We will also add the details and results based on state-of-the-art dense prediction frameworks. Due to the page limit, we have included these new results and discussions in the revised appendix (supplementary material), where we highlight the modified parts in blue.
>
> [A] Hybrid task cascade for instance segmentation, CVPR 2019
>
> [B] DINO: DETR with Improved DeNoising Anchor Boxes for End-to-End Object Detection, arXiv:2203.03605
>
> [C] Masked-attention mask transformer for universal image segmentation, CVPR 2022
>
> [D] Objects365: A Large-scale, High-quality Dataset for Object Detection, ICCV 2019
>
> [E] COCO-Stuff: Thing and Stuff Classes in Context, CVPR 2018
>
> [F] Vision Transformer Adapter for Dense Predictions, ECCV 2022

---

> > ### Author Response · Authors · 2022-08-02
> > **Response to Reviewer j8qU, Part II**
> >
> > > **Q2: Throughput of each method**
> >
> > **[Reply]** Thanks very much for this suggestion.  We have compared the latency on GPU with the two main baseline methods in Figure 3(c). We agree that the multiple small matrix multiplications introduced by our method will affect the speed of our method on GPU. We also observed that our method is slower than ConvNeXt by 7%~15% with similar FLOPs. Meanwhile, thanks to the highly efficient depth-wise convolutions implementation of CuDNN, we also see that our models achieve similar or slightly faster speeds than typical vision Transformers with similar FLOPs. Notably, as shown in Figure 3(c), the higher classification accuracy helps our models achieve better speed-accuracy trade-offs than ConvNeXt and Swin Transformers. Therefore, we believe the speed of our method is still competitive with these recent models.  We provide the detailed throughput statistics in the following table. Apart from ConvNeXt and Swin Transformers, we also include more powerful MViTv2-T/S/B models as suggested by the reviewer (since the other two models are not publicly available, we cannot measure their throughput). We have added these results in the revised appendix (supplementary material), where we highlight the modified parts in blue.
> >
> >
> > |Model | GFLOPs| Throughput (images / s) | Acc. (%)|
> > |------|-------|--------|-------|
> > |ConvNeXt-T| 4.5 |   1010.3   |82.1 |
> > |Swin-T| 4.5 |   832.2   |81.3 |
> > |MViTv2-T| 4.7 |   728.4   |82.3 |
> > |HorNet-T$_{7\times 7}$| 4.0 |   845.7   |82.7 |
> > |ConvNeXt-S| 8.7 |   621.5   |83.1 |
> > |Swin-S| 8.7 |   520.7   |83.0 |
> > |MViTv2-S| 7.0 |  531.5   |83.6 |
> > |HorNet-S$_{7\times 7}$| 8.8 |   525.8   |83.8 |
> > |ConvNeXt-B| 15.4 |  440.8   |83.8 |
> > |Swin-B| 15.4 |   364.8   |83.5 |
> > |MViTv2-B| 10.2 |  369.1   |84.4 |
> > |HorNet-B$_{7\times 7}$| 15.6 |   410.0   |84.2 |
> >
> >
> > >**Q3: Theoretical analysis of high-order interactions**
> >
> > **[Reply]** Thanks very much for this suggestion. We agree that theoretical analysis of the proposed high-order interaction mechanism is helpful to better understand our model. But to be honest, currently, we could not find a good theorem to explain such interaction mechanisms in deep networks, since theoretically analyzing a complex system like HorNet or $\textit{g}^\textit{n}\text{Conv}$ is very difficult. To the best of our knowledge, there is also no theorem to thoroughly analyze the effectiveness of the prevalent self-attention mechanism. Therefore, we would like to leave this as future work. In our paper, we have some empirical and intuitive analyses to show the effectiveness of the high-order interaction mechanism in Section 3.1 and Appendix B. We hope these explanations can help readers to better understand our motivation and provide some guidance to design better architectures in future research.

---

> > > ### Author Response · Authors · 2022-08-07
> > > **Looking forward to your feedback**
> > >
> > > Dear reviewer j8qU,
> > >
> > > Does our response address your concerns? Please feel free to let us know if you have any further questions.
> > >
> > > Best wishes!

---

### Official Review · Reviewer_dnYU · 2022-07-11

**Rating:** 5
**Confidence:** 4
**Soundness:** 3 good
**Presentation:** 3 good
**Contribution:** 3 good

**Summary:**

The paper presents the Recursive Gated Convolution (g^n Conv) that performs high-order spatial interactions with gated convolutions and recursive designs. The proposed module can serve as a plug-and-play module for both transformer and convolutional neural networks.
Based on the proposed module, HorNet is further introduced, showing good results on ImageNet classification, COCO object detection and ADE20K semantic segmentation.



**Questions:**

See the Strengths And Weaknesses part

**Limitations:**

See the Strengths And Weaknesses part

**Strengths And Weaknesses:**


Pros:
- Well organized and good writing.
- Extensive experiments on multiple datasets and tasks.
- The design of g^nConv is interesting and contributes much to the performance.
- Limitations are stated in this paper.


Cons:
- The motivation is not very clear to me. Why do we need high-order interactions?
- Since the operator is Mul rather than MatMul, then the high-order interaction seems like an element-wise one rather than the spatial-wise interaction as stated in the paper. Or it can be recognized as a channel-wise one (element-wise multiply -> channel projection -> element-wise multiply ...)?
- Could the author provide the complexity analysis for different n?

---

> ### Author Response · Authors · 2022-08-02
> **Response to Reviewer dnYU**
>
> We sincerely thank the reviewer for the detailed comments and advice. We address the questions and clarify the issues accordingly as described below.
>
> >**Q1: About the motivation of high-order interactions**
>
> **[Reply]** Our motivation originates from the observation of the recent success of vision Transformers. We identify the three key ingredients behind the success of recent vision Transformer models: input-adaptive, long-range, and high-order spatial interactions. While previous work borrows new designs including large kernels and input-adaptive weights, we demonstrate that the explicit 2-order spatial interactions achieved by the self-attention operation (as shown in Fig. 1) are beneficial for vision models. Therefore, it is natural to investigate whether higher-order interaction can further enhance the modeling capacity of vision models.
>
> >**Q2: About the spatial-wise interaction**
>
> **[Reply]** Our high-order interaction operator ($\textit{g}^\textit{n}\text{Conv}$) is designed to be spatial-wise instead of element-wise. Although $\textit{g}^\textit{n}\text{Conv}$ is built with only depth-wise convolution, linear projection and element-wise multiplication, we have shown that $\textit{g}^\textit{n}\text{Conv}$ can indeed achieve high-order spatial interactions efficiently. The recursive formula of $\textit{g}^\textit{n}\text{Conv}$ is $p_{k+1}=f_k(q_k)\odot g_k(p_k)$  (see Equ. (3.3)),
> where $f_k$ is a depth-wise convolution and $g_k$ is a linear projection or identity mapping.
> Therefore, we have
> $$
> p_{k+1}^{(i, c)}=\sum_{j\in\Omega_i} w_{i\to j}^c q_k^{(j,c)}g_k^{(i, c)},
> $$
> where $\Omega_i$ denotes the receptive field of $f_k$ and we show that the feature at spatial location $i$ explicitly interacts with another feature at spatial location $j$. As a result, each recursive step will increase the order of spatial interaction and the final output of $\textit{g}^\textit{n}\text{Conv}$ considers $n$-order spatial interactions.
>
> To sum up, the depth-wise convolution aggregates features from different spatial locations while the element-wise multiplication helps to introduce explicit interaction. To better understand why our operation introduces spatial interactions rather than channel-wise interactions, please also refer to the analysis in Equ. (3.7) and (3.8), where we show our $\textit{g}^\textit{n}\text{Conv}$ can accomplish the goal of input-adaptive spatial mixing like self-attention.
>
> >**Q3: About the complexity analysis of different $n$**
>
> **[Reply]** Thanks for your suggestion. The complexity of our $\textit{g}^\textit{n}\text{Conv}$  has an upper bound ${\rm FLOPs}(\textit{g}^\textit{n}\text{Conv}) < HWC(2K^2 + 11/3\times C + 2)$ (see Equ. (3.6)), thanks to the design of channel dimension for each order
> $$
>   C_k = \frac{C}{2^{n-k-1}}, \qquad 0\le k\le n-1.
> $$
> We have also provided a closed form of the complexity in Appendix A:
> \begin{equation}
>     {\rm FLOPs}(\textit{g}^\textit{n}\text{Conv}) = HWC\left[2K^2\left(1 - \frac{1}{2^n}\right) + \left(\frac{11}{3} - \frac{2}{3\times 4^{n-1}}\right)C + 2 - \frac{1}{2^{n-1}}\right],
> \end{equation}
> where one can easily derive the upper bound of the complexity of our $\textit{g}^\textit{n}\text{Conv}$.

---

> > ### Author Response · Authors · 2022-08-07
> > **Looking forward to your feedback**
> >
> > Dear reviewer dnYU,
> >
> > Does our response address your concerns? Please feel free to let us know if you have any further questions.
> >
> > Best wishes!

---

> > > ### Comment · Reviewer_dnYU · 2022-08-07
> > > **Post-rebuttal comments**
> > >
> > > Thanks for the response, which solves my questions well. I still need precision on motivation in order to adjust my final rate.
> > >
> > > "We identify the three key ingredients behind the success of recent vision Transformer models: input-adaptive, long-range, and high-order spatial interactions." How to understand "high-order spatial interactions" here? Could I request a more concrete explanation?
> > > Thank you

---

> > > > ### Author Response · Authors · 2022-08-07
> > > > **Thanks for your feedback**
> > > >
> > > > Thanks a lot for checking our response and providing valuable feedback.
> > > >
> > > > High-order spatial interaction is the key concept introduced in our paper. Previous work on Transformer-like architectures usually explores the long-term and input-adaptive weights in the self-attention mechanism. In this paper, we find the high-order spatial interactions in the self-attention mechanism are also critical to the expressive power of Transformers.
> > > > For vision Transformers, the high-order spatial interactions in our paper represent the two-order interactions among $\mathbf{q}$, $\mathbf{k}$, and $\mathbf{v}$. In each self-attention layer, the input feature $\textbf{x}$ is first transformed into three versions $\mathbf{q}$, $\mathbf{k}$ and $\mathbf{v}$ via linear projections. The attention weight of position $i$ is computed by $\mathbf{a}_i=\mathbf{q}_i^{\top}[\mathbf{k}_1,\ldots,\mathbf{k}_n]$,
> > > > which is the first spatial interaction since the relations between $\mathbf{q}_i$ and $[\mathbf{k}_j, j=1,\ldots,n]$ from all $n$ spatial locations are considered. The attention weights then interact with $\mathbf{v}$ to obtain the final output: $\mathbf{x}_i = \sum_j\hat{\mathbf{a}}_i\mathbf{v}_j$, which is the second spatial interactions since the relations of $\mathbf{a}$ and $\mathbf{v}_j$ from all spatial locations are considered. Since the second interaction depends on the first one, we regard the whole process as a two-order interaction among different spatial locations.
> > > >
> > > > As shown in Figure 1, we extend the above analysis to other popular operations in deep vision models. It is worth noting that while there exists complex and often high-order interactions between two spatial locations in a deep model due to the non-linearity, we find that the **explicit** and **high-order spatial interactions** introduced by the architectural designs are beneficial to improving the modeling power of vision models. Therefore, we further explore the concept of high-order interactions and propose our recursive gated convolutions which can accomplish arbitrary-order spatial interactions with a highly efficient implementation. Our ablation study in Table 4(a) also shows that the performance of models can be largely improved when we introduce high-order spatial interactions. We have also provided a more formal analysis of high-order interactions in line 179-183 and Appendix B.
> > > >
> > > > We hope our response can address your concerns. Please let us know if you have further questions on this issue.

---

> > > > > ### Comment · Reviewer_dnYU · 2022-08-07
> > > > > **Thanks again for the explanation**
> > > > >
> > > > > Thanks for addressing my concerns. I have increased the rating to 5.

---

### Author Response · Authors · 2022-08-02
**General Response to All Reviewers**

Dear reviewers and area chair,

First, we would like to thank you all for your time and insightful comments on our paper. We are encouraged to hear that the reviewers think our design is interesting and contributes much to the performance (Reviewer dnYU), our architecture is novel (Reviewer j8qU), simple and effective (Reviewer 3Snz), the writing is good and clear (all reviewers), our experiments are extensive (Reviewer dnYU), carefully designed (Reviewer j8qU), thorough and solid (Reviewer 3Snz).

We also appreciate their suggestions to improve our work. We notice both reviewers j8qU and 3Snz raise concerns about the speed of our models. We provide a throughput analysis in our rebuttal. Although our models are slower than ConvNeXt series, we find HorNet can generally achieve similar or slightly fast speeds than various vision Transformers. Notably, as shown in Figure 3(c), the higher classification accuracy helps our models achieve better speed-accuracy trade-offs than ConvNeXt and prevalent vision Transformers. Therefore, our models still exhibit very competitive complexity-accuracy trade-offs with these recent state-of-the-art methods.

Since our experiments are designed to verify the superior of our new operation over previous basic operations like plain convolution and self-attention, we directly adopt the overall architectures and training configurations of ConvNeXt and Swin Transformers. As mentioned by reviewer j8qU, this experiment setup leads to lower performance on ImageNet-1K compared to some recent methods. In our response, we list several possible directions to further improve our models. Besides, we also show that HorNet can achieve state-of-the-art level performance on downstream object detection and semantic segmentation by directly adopting several recent dense prediction frameworks thanks to our simple architecture.

We also include discussions on the related papers suggested by the reviewers. Due to the page limit, we include the additional analysis and discussions in the appendix, where we highlight the modified parts in blue.

We thank you again for your time and feedback. We hope our response can address your concerns.

---

### Meta-Review · Area_Chair_uMga · 2022-08-29

**Recommendation:** Accept
**Confidence:** Certain

**Metareview:**


This paper introduces a new operation **gnConv** and a computer vision network architecture **HorNet**.  Motivated by the success philosophy of vision Transformers, the key idea of gnConv is to build a recursive form of gated convolution. It make the module input-adaptive and with long-range and high-order spatial interactions. Consistent improvement are shown over Swin and ConvNeXt on well-established CV benchmarks such as image classification on ImageNet, semantic segmentation on ADE20K and object detection on COCO.

The paper receives receives unanimous accept from all reviewers (Reviewer 3Snz champions the paper with rating score 8), leading to an ``Accept'' decision.

**Award:**

No

---

### Decision · Program_Chairs · 2022-09-14

Accept